# ReLiFiRa (Real Life Filgotinib in Rheumatoid Arthritis): Retrospective Study of Efficacy and Safety in Common Clinical Practice

**DOI:** 10.3390/jpm13091303

**Published:** 2023-08-25

**Authors:** Maurizio Benucci, Marco Bardelli, Massimiliano Cazzato, Elenia Laurino, Francesca Bartoli, Arianna Damiani, Francesca Li Gobbi, Anna Panaccione, Luca Di Cato, Laura Niccoli, Bruno Frediani, Marta Mosca, Serena Guiducci, Fabrizio Cantini

**Affiliations:** 1Rheumatology Unit, San Giovanni di Dio Hospital, 50143 Florence, Italy; francesca.ligobbi@uslcentro.toscana.it; 2Rheumatology Unit, Department of Medicine, Surgery and Neurosciences, University of Siena, 53100 Siena, Italy; mar.cobar@hotmail.it (M.B.); fredianibruno60@gmail.com (B.F.); 3Unit of Rheumatology, University Hospital of Pisa, 56126 Pisa, Italy; maxxmed@live.com (M.C.); elenia.laurino@gmail.com (E.L.); marta.mosca@unipi.it (M.M.); 4Department of Clinical and Experimental Medicine, University of Florence, 50134 Florence, Italy; francesca.bartoli19@gmail.com (F.B.); ariannadam@outlook.it (A.D.); serena.guiducci@unifi.it (S.G.); 5Internal Medicine and Rheumatology Unit, Santa Maria General Hospital, 05100 Terni, Italy; anna.panaccione@tiscali.it (A.P.); l.dicato@aospterni.it (L.D.C.); 6Division of Rheumatology, Prato Hospital, 59100 Prato, Italy; laura.niccoli@uslcentro.toscana.it (L.N.); fbrzcantini@gmail.com (F.C.)

**Keywords:** rheumatoid arthritis, filgotinib, real-life

## Abstract

Background: Filgotinib (FIL) is a selective JAK1 inhibitor with an affinity 30-fold higher than JAK2, approved to treat moderate to severe active rheumatoid arthritis (RA), in adults with inadequate response or intolerance to one or more disease-modifying antirheumatic drugs (DMARDs). Methods: We conducted a retrospective, multicentric study in order to evaluate efficacy and safety of FIL 200 mg daily therapy, after 3 and 6 months, in 120 patients affected by RA, managed in Tuscany and Umbria rheumatological centers. The following clinical records were analyzed: demographical data, smoking status, previous presence of comorbidities (Herpes zoster -HZ- infection, venous thromboembolism -VTE-, major adverse cardiovascular events -MACE-, cancer, diabetes, and hypertension), disease duration, presence of anti-citrullinated protein antibodies (ACPA), rheumatoid factor (RF), number of biological failures, and prior csDMARDs utilized. At baseline, and after 3 (T3) and 6 (T6) months of FIL therapy, we evaluated mean steroid dosage, csDMARDs intake, clinimetric indexes (DAS28, CDAI, HAQ, patient and doctor PGA, VAS), erythrocyte sedimentation rate (ESR), C-reactive protein (CRP), and body mass index (BMI). Results: At baseline, the mean disease duration was 9.4 ± 7.5 years; the prevalence of previous HZ infection, VTE, MACE, and cancer was respectively 4.12%, 0%, 7.21%, and 0.83%, respectively. In total, 76.3% of patients failed one or more biologics (one biological failure, 20.6%; two biological failures, 27.8%; three biological failures, 16.5%; four biological failures, 10.3%; five biological failures, 1.1%). After 3 months of FIL therapy, all clinimetric index results significantly improved from baseline, as well as after 6 months. Also, ESR and CRP significatively decreased at T3 and T6. Two cases of HZ were recorded, while no new MACE, VTE, or cancer were recorded during the observation time. Conclusion: Despite the limitations of the retrospective study and of the observational period of only 6 months, real-life data on the treatment of RA patients with FIL demonstrate that this Jak inhibitor therapy is safe in terms of CV, VTE events, and occurrence of cancer, and is also effective in a population identified as “difficult to treat” due to failure of previous b-DMARD therapy.

## 1. Introduction

Since the approval of the first-generation of Janus kinase (JAKi) inhibitors, tofacitinib and baricitinib, the search for new innovative JAK inhibitors with more specific selectivity has started. It has been hypothesized that the inhibition of JAK1 will allow the same clinical efficacy as a non-selective pan-JAK inhibitor (or event better as the dose could be increased), but with a better safety profile potentially guaranteed by the non-inhibition of JAK2-JAK3 [1]. Filgotinib (FIL) is a selective JAK inhibitor with a selectivity for JAK1 vs. JAK2 of nearly 30-fold [2]. The core clinical program evaluating FIL in patients with moderately-to-severely active rheumatoid arthritis (RA) consists of three phase 2b (DARWIN 1–3) and four phase 3 (FINCH 1–4) studies. In accordance with the standard clinical use of targeted therapies, the development scheme of FIL mainly focused on its use as combination therapy with methotrexate (MTX) or other conventional synthetic disease-modifying antirheumatic drugs (csDMARDs) in three patient populations defined by treatment history, although FIL monotherapy was also evaluated in some studies. DARWIN 1, DARWIN 3, and FINCH 1 studies evaluated the use of FIL in combination with MTX in patients with IRs to MTX as per the second-line therapy recommended in the EULAR treatment algorithm. FINCH 2 evaluated the use of FIL in combination with csDMARDs in patients with prior bDMARD failure or intolerance, i.e., as per the third-line therapy recommended in the EULAR treatment algorithm. FINCH 3 evaluated the use of FIL in MTX-naïve patients. Data are not yet available for the ongoing long-term extension, the FINCH 4 study [3]. A recent trial evaluated 1455 patients who received tofacitinib (TOFA) at a dose of 5 mg twice daily, 1456 received TOF at a dose of 10 mg twice daily, and 1451 received a TNF inhibitor. During a median follow-up of 4.0 years, the incidence of major adverse cardiovascular events (MACE) and cancer was higher with the combined doses of TOFA (3.4% (98 patients) and 4.2% (122 patients), respectively) compared with to a TNF inhibitor (2.5% (37 patients) and 2.9% (42 patients), respectively). The hazard ratios were 1.33 (95% confidence interval (CI), 0.91–1.94) for MACE and 1.48 (95% CI, 1.04–2.09) for tumors; non-inferiority of TOFA has not been demonstrated [4]. This has led the CHMP-EMA to class warnings for TOFA, baricitinib (BARI), upadacitinib (UPA), and FIL [5]. Real-life data from the STAR-RA study [6], the post hoc analysis of the ORAL study in patients under 65 years of age and with low risk for ASCVD < 5% (atherosclerotic cardio-vascular disease) [7], and data from the TOFA rheumatoid arthritis clinical program showed absence of this risk in patients treated with TOFA [8]. A pharmacovigilance study analyzing and evaluating the association between thromboembolic events and the use of JAKinib, based on the latest data from the Food and Drug Administration’s Adverse Event Reporting System provided new safety signals based on previous event information thromboembolic [9]. Real-world evidence on JAKis is primarily available for TOFA, from countries where it was introduced earlier. Although generally reporting similar effectiveness between tofacitinib and bDMARDs [10,11], two large studies have suggested improved drug persistence on tofacitinib compared with TNFis, at least after failure of a first biological DMARD (bDMARD) [12,13]. Real-world evidence remains limited for BARI, which, so far, has mainly been compared with TOFA in small studies with limited ability to control for confounding [14,15]. The aim of our research was to retrospectively collect the efficacy and safety data of 120 rheumatoid arthritis patients mostly refractory to other b-DMARD therapies treated with FIL 200 mg/day, followed in rheumatological centers in Tuscany and Umbria.

## 2. Materials and Methods

The efficacy and safety of filgotinib 200 mg daily therapy for 3 and 6 months was evaluated in 120 RA patients observed. We retrospectively collected the data of three evaluations three months apart of 120 rheumatoid arthritis patients followed up in the rheumatological centers in common clinical practice in the period June–December 2022. The following clinical records were analyzed: demographical data, smoking status, previous presence of comorbidities (Herpes zoster -HZ- infection, venous thromboembolism -VTE-, major adverse cardiovascular events -MACE-, cancer, diabetes, and hypertension) disease duration, presence of anti-citrullinated protein antibodies (ACPA), rheumatoid factor (RF), number of biological failures, and prior csDMARDs utilized. At baseline, and after 3 (T3) and 6 (T6) months of FIL therapy, we evaluated mean steroid dosage, csDMARDs intake, clinimetric indexes (DAS28, CDAI, HAQ, patient and doctor PGA, VAS), erythrocyte sedimentation rate (ESR), C-reactive protein (CRP), and body mass index (BMI). We also evaluated some laboratory parameters of tolerability (total cholesterol (TC), cholesterol LDL (C-LDL), cholesterol HDL (C-HDL), triglycerides, creatinines, aspartate aminotransferase (AST), alanine aminotransferase (ALT), and hemoglobin). The study involving human participants was reviewed and approved by the GISEA Project Ethics Review Board on 22 September 2020 (Code of Ethics 6496_OSS). Written informed consent was not required for participation in this study in accordance with national legislation and institutional requirements.

## 3. Statistical Analysis

Descriptive statistics have been used to describe the basic features of the population and, because the data did not show a normal distribution, we utilized median and Inter Quartile Range (IQR). For comparisons, the Wilcoxon test for paired sample was used. A *p*-value less than 0.05 was considered statistically significant. Statistical analysis was performed by © 2023 MedCalc Software Ltd., Acacialaan 22, 8400 Ostend Belgium.

## 4. Results

Table 1 shows the baseline clinical and demographic characteristics of the patients included into the study. At baseline, the percentage of females was 85.5% and males was 14.5%, mean age was 62.58 ± 14.64, and 54.6% of patients were ≥65-year-old. Smoking was present in 18.5%, hormone replacement therapy in 1.03%, prior Herpes zoster infection was 4.12%, prior venous thrombo-embolism (VTE) 0%, prior major adverse cardiovascular events (MACE) 7.21%, prior cancer 0.83%, diabetes 15.46%, and hypertension 47.42%, and the mean duration of disease was 9.4 ± 7.5 years. It is important to highlight that 63% of the patients had cardiovascular risk factor, in particular, 38% had one risk factors, 23% two risk factor, and 2% three risk factors. In total, 76.3% of patients failed biologic therapy (one biological failure, 20.6%; two biological failures, 27.8%; three biological failures, 16.5%; four biological failures, 10.3%; five biological failures, 1.1%). Of the 91 patients who had failed a previous b-DMARD, 25 patients had one failure (20 anti-TNF, 3 Tocilizumab (TOC), 2 Abatacept (ABA)), 33 patients had three failures (33 anti-TNF, 18 TOC, 15 ABA), 20 patients had four failures (20 anti-TNF, 20 TOC, 20 ABA), 12 patients had four failures (12 anti-TNF, 12 TOC, 12 ABA, 12 Rituximab (RTX)), and 1 patient had five failures (1 anti-TNF, 1 TOC, 1 ABA, 1 Sarilumab (SAR), 1 RTX). In the observed population, no patients had been previously treated with other targeted synthetic disease-modifying anti-rheumatic drugs (ts-DMARDs). Regarding autoantibodies, anti-citrullinated protein (ACPA) was present in 80.4% of patients, rheumatoid factor (RF) in 91.75%, and double positive was detectable in 79.38% of the population. Regarding cDMARDs, methotrexate was used in 35% of patients, leflunomide in 5.15%, and sulfasalazine in 4.12%. Statins were used in 12.37% of patients, mean steroid dosage was 4 (0–7.7) mg/day. Regarding clinical and laboratories evaluation, Disease activity assessed by DAS28 was 4.76 (4.34–5.55), CDAI was 21 (18–22.75), Erythrocyte sedimentation rate (ESR) was 35 (18–48) mm/h, C-reactive protein (CRP) was 1.18 (0.5–1.89) mg/dL, tender joint count was 6 (5–11), swollen joint count was 4 (4–8), VAS 7 (5–8), HAQ 1 (1–1.5), patient PGA 70 (40–80), physician PGA 65 (30–70), mean body weight was 68 (58–75), and BMI 23.15 (20.79–26.30).

After 3 and 6 months of treatment with FIL (as showed in Table 2), it is possible to observe a statistically significant reduction in steroid dosage from 4 (0–7.7) mg at baseline to 0 (0–4) mg at month 3, and then up to 0 (0–4) mg at month 6 (T0 vs. T3 *p* = 0.0001; T0 vs. T6 *p* = 0.0001; T3 vs. T6 *p* = 0.048). The mean value of DAS28 decreases from 4.76 (4.34–5.55) at baseline to 3.03 (2.67–3.63) at 3 months, reaching 2.7 (2.29–3.1) after 6 months (T0 vs. T3 *p* = 0.0001; T0 vs. T6 *p* = 0.0001; T3 vs. T6 *p* = 0.0001). The CDAI result is very similar, decreasing from 21 (18–22.75) at baseline to 13 (12–15) in the third month, and down to 9 (6–10.25) at the end of the observation. (T0 vs. T3 *p* = 0.0001; T0 vs. T6 *p* = 0.0001; T3 vs. T6 *p* = 0.0001). ESR and CRP show the same evolution: at baseline, the ESR was 35 (18–48) mm/h, at the third month it was 15 (10–29.75) mm/h, and at the sixth month it was 13 (10–28) mm/h (T0 vs. T3 *p* = 0.0001; T0 vs. T6 *p* = 0.0001; T3 vs. T6 *p* = 0.026); CRP at baseline was 1.18 (0.5–1.89) mg/dL; after three months, it was 0.47 (02–0.85) mg/dL; and at 6 months, it was 0.28 (0.12–0.49) mg/dL (T0 vs. T3 *p* = 0.0001; T0 vs. T6 *p*= 0.0001; T3 vs. T6 *p* = 0.0001). Tender and swollen joint counts show a substantial reduction: at baseline, tender joints were 6 (5–11); at three months, they were 2 (1–3); and at month 6, they were 1 (0–2) (T0 vs. T3 *p* = 0.0001; T0 vs. T6 *p* = 0.0001; T3 vs. T6 *p* = 0.0001); swollen joint count was 4 (4–8) at baseline, 1 (1–2) at month 3, and 0 (0–1) at month 6 (T0 vs. T3 *p* = 0.0001; T0 vs. T6 *p* = 0.0001; T3 vs. T6 *p* = 0.0001). The result of the VAS pain value decreased from 7 (5–8) at baseline, to 2.5 (1–5) at month 3, to 2 (1–3) at month 6 (T0 vs. T3 *p* = 0.0001; T0 vs. T6 *p* = 0.0001; T3 vs. T6 *p* = 0.0002), as well as functional scale HAQ which started from 1 (1–1.5) at baseline and reached 0.25 (0–0.5) at month 3 remaining stable at the same value, 0.5 (0–0.5), up to the sixth month (T0 vs. T3 *p* = 0.0001; T0 vs. T6 *p* = 0.0001; T3 vs. T6 NS). Regarding patient and physician assessment, the results are consistent: patient PGA at baseline was 70 (40–80); at three months, it was 27.5 (15–41.25); and at six months, it was 20 (10–30) (T0 vs. T3 *p* = 0.0001; T0 vs. T6 *p* = 0.0001; T3 vs. T6 *p* = 0.0001). Regarding the PGA physician, the baseline value was 65 (30–70) at three months 20 (10–45) and at six months, it was 20 (10–30) (T0 vs. T3 *p* = 0.0001; T0 vs. T6 *p* = 0.0001; T3 vs. T6 *p* = NS). After three and six months, no statistically significant differences were observed in the lipid profile or in renal or hepatic function. Regarding adverse events, only two cases of Herpes zoster were observed in the first three months, while no cases of VTE and MACE were recorded in six months of treatment with FIL.

## 5. Discussion

In light of current knowledge, this is the first real-life study of FIL treatment in rheumatoid arthritis in a population identifiable as difficult-to-treat (D2T-RA) due to the high proportion of patients who had failed prior b-DMARD therapy. Our patients had failed previous biological therapies in 76.3% of the population (one biological failure, 20.6%; two biological failures, 27.8%; three biological failures, 16.5%; four biological failures, 10.3%; five biological failures, 1.1%). FINCH 2 evaluated the use of FIL in combination with csDMARDs in patients with prior bDMARD failure or intolerance, i.e., as per the third-line therapy recommended in the EULAR treatment algorithm. In this 24-week phase 3 study, patients with IRs or intolerance to >1 prior bDMARD were randomized to receive placebo, FIL 200 mg, or FIL 100 mg. Results of primary and key secondary endpoints supported the superior efficacy of both FIL doses vs. placebo. Subgroup analyses showed that the efficacy of FIL was not affected by the number or mechanism of action (MOA) of prior bDMARDs, as patients with >3 prior bDMARDs or >1 MOA of prior bDMARDs, as well as those previously exposed to IL-6 inhibitors or TNFis, all achieved efficacy outcomes comparable to the overall study population [16,17,18]. Despite the failure of previous therapies with b-DMARDs, the patients treated with Jak-inhibitor showed a rapid improvement in clinimetric (DAS28, CDAI) functional parameters (HAQ, VAS, PhGA) in line with studies that highlighted the ability of this kind of drugs to act on D2T-RA [19]. According to FIRST registry where 353 D2T-RA and 106 very-D2T-RA patients (failure of 2–3 b-DMARDs) JAKi showed significant improvement in CDAI in D2T RA and vD2T RA patients, compared with reverse probability treatment weighted (IPTW). Latent class analysis of treatment response trajectories revealed that the proportion of a group of patients who showed poor response was lower in the JAKi subgroup than in those with other subgroups [19]. This ability of Jak-inhibitors in D2T-RA is probably due to their action on the different pathogenetic phases of RA both on innate and adaptive immunity, T cell mediated [20], and the ability to interfere on B lymphocytes in the antibody production [21]. An interesting aspect of our observation was that 13 patients (12 as fourth line of treatment, 1 as fifth line of treatment) had already undergone cycles of therapy with RTX. In our study, in line with the EULAR Recommendations [22], we observed a statistically significant reduction in corticosteroid dosage, a condition which represents an important key-point in the definition of the treatment’s success in D2T-RA patients [23]. Another interesting aspect of our study is that the data collection began before the most recent EULAR recommendations about the treatment with Jak-inhibitors [22]. Our patients were, on average, 62 years old, so they are below the threshold of age 65 indicated by EULAR as risk factor for Jak-inhibitor treatment. However, 54.6% of patients were ≥65-year-old and this condition could be a feature able to drive, according to the EULAR recommendations, an increase in adverse event. Moreover, there were other important safety aspects that we have to consider: smokers represented 18.5% of the population, 1.03% were treated with hormone replaces therapy, prior MACE was recorded in 7.21% of the patients, prior cancer in 0.83%, diabetes in 15.46%, and hypertension in 47.42% of the population under investigation. Despite the presence of risk factors, in the 6 months of observation, no new cardiovascular, thrombo-embolic events were observed. A limitation of our observation carried out in the year 2022 was also that of not evaluating in the population of patients with RA the Italian recommendations for the assessment of cardiovascular risk in rheumatoid arthritis and the recent position paper of the Cardiovascular Obesity and Rheumatic DISease (CORDIS) Study Group of the Italian Society of Rheumatology which evaluates the baseline risk according to the CUORE chart (https://www.cuore.iss.it/valutazione/calc-rischio, access on 12 May 2023) [24]. Real life data from STAR-RA demonstrated that the pooled weighted HR (95% CI) for CV outcomes when comparing TOFA with TNFi was 1.01 (0.83 to 1.23) with weighted rate difference (95% CI) corresponding to 0.02 (−0.19 to 0.23) CV events per 100 person-years [6]. Additionally, post hoc analysis of the ORAL surveillance study [7] and data from the TOFA rheumatoid arthritis clinical program [8] demonstrated no CV risk for those patients with low or moderate atherosclerosis cardiovascular disease (ASCVD) at baseline. In our group of patients, although only the 12.37% were treated with statins at baseline, in six months of observation, we did not observe any increase in total cholesterol (TC), cholesterol-LDL(C-LDL), or triglycerides (TG). These data arriving from a real observation probably differentiate FIL from other Jak inhibitors. These kinds of drugs not only block cell signaling via JAK/STAT, but they have also cell metabolic effects (including decreased mitochondrial membrane potential, mitochondrial mass, and ROS and inhibition of metabolic genes in synovial tissue) [25] and they are able to modify the systemic lipid metabolism. TOFA and BARI significantly increased C-HDL and C-LDL after treatment, compared with baseline and with other DMARD, as shown in RA randomized controlled trials [26,27,28], an effect that is possible to revert by statins [26]. JAK inhibitors also improve HDL function by increasing the activity of lecithin–cholesterol acyltransferase (LCAT; an enzyme that converts free cholesterol to cholesterol esters and supports cholesterol efflux to lipoproteins), increasing HDL efflux capacity [26,27]. Other effects, such as alterations in lipoprotein size and content, have been described [29,30]. Although the treatment with UPA increases both C-LDL and C-HDL levels, it had no significant effect on the cardiovascular risk during a 52-week follow-up [31]. A recent systematic review and network meta-analysis has been performed about randomized controlled trials in RA from Pubmed, Medline, Embase, and the Cochrane Controlled Trials Register. The primary outcome was mean change in C-HDL e C-LDL from baseline. The mean treatment differences and range of effect of various JAKi on C-HDL and C-LDL were estimated. Based on data from 18 unique studies involving five approved JAK inhibitors and 6697 RA patients (JAKi = 3341, placebo = 3356), these inhibitors led to a mean increase of 8.11 mg/dL in HDL levels from baseline and a mean increase of 11.37 mg/dL in LDL levels from baseline. Cardiovascular disease risk did not differ significantly between patients who received JAK inhibitors or those who received placebo or active agents [32]. An important limitation of our study for lipid metabolism is represented by the short duration of the observation, only six months, which is, perhaps, insufficient to notice differences on the levels of TC, C-LDL, and C-HDL [27,30]. The better selectivity of FIL on Jak-1 in the absence of activity on Jak-2 could determine an absence of action on leptin, maintaining a stable satiety and that the action on the lipid profile is only mediated by the inflammatory mechanism IL-6 mediated such as Tocilizumab [33,34]. Currently our study represents the only real-life research on the Italian population with D2T-RA treated with FIL. Previously, a study prospectively enrolled 446 BARI-treated rheumatoid arthritis patients from 11 Italian centers and evaluated at baseline and after 3, 6, and 12 months. They were classified based on prior treatments as bDMARD-naïve and bDMARD-insufficient responders (IRs) after bDMARD intolerance or failure. A subanalysis differentiated the effects of methotrexate (MTX) and oral glucocorticoid (OGC) use. The cohort included 150 (34%) bDMARD-naïve patients and 296 (66%) bDMARD-IR patients, including 217 (49%) using baricitinib monotherapy. Considering DAS-28-CRP as the primary outcome, at 3 and 6 months, 114/314 (36%) and 149/289 (51.6%) patients achieved remission, respectively, while those with low disease activity (LDA) represented 62/314 (20%) and 46/289 (15.9%), respectively. Finally, at 12 months, 81/126 (64%) were in remission and 21/126 (17%) in LDA. As in our series, a significant reduction in OGC dose was observed at 3 and 12 months in all groups. Fifty-eight (13%) patients discontinued baricitinib due to adverse events, including thrombotic events and herpes zoster reactivation [35]. A second study was conducted to evaluate the retention rate in 23 Italian tertiary rheumatological centers. The study considered a treatment period of up to 48 months for all included patients. The retention rate of TOFA was evaluated with the Kaplan–Meier method. Hazard ratios (HRs) for TOFA discontinuation were obtained using Cox’s regression analysis. Analysis of data from 213 patients revealed that the TOFA retention rate was 86.5% (95% CI: 81.8–91.5%) at month 12, 78.8% (95% CI: 78.8–85.2%) at month 24, 63.8% (95% CI: 55.1–73.8%) at month 36, and 59.9% (95% CI: 55.1–73.8%) at month 48 after initiation of treatment. None of factors analyzed, including number of prior treatments received, disease activity or duration, presence of RF and/or ACPA, and presence of comorbidities, were predictive of TOFA retention rate [36]. A third recent study retrospectively evaluated rheumatoid arthritis patients who received a JAKi (TOFA, BARI, UPA, FIL) from four tertiary care centers in Milan. Six-hundred-and-eighty-five patients were included and received BARI (48%), TOFA (31%), UPA (14%), or FIL (7%), with 47% as an innovative first-line treatment before a biologic. Out of a total of 1137 patient-years, there was 1 stroke and 123 (18%) adverse events of special interest (AESI), including 3 deaths, all due to serious infections. Among patients with a higher cardiovascular risk, a higher frequency of adverse events of special interest was observed (23%) [37].

## 6. Conclusions

Despite the limitations of the retrospective study and of the observational period of only 6 months, data on the treatment of RA patients with FIL demonstrate that this Jak inhibitor therapy is safe in terms of CV and VTE events, and is also effective in a population identified as “difficult to treat” due to failure of previous b-DMARD therapy. Real-life and long-term observational studies in the light of the CHMP-EMA recommendations [5] and a prior selection of patients according to the recommendations of the Italian Society of Rheumatology [24] are necessary to mitigate doubts about ts-DMARDs and differences for these therapies they represent in terms of efficacy a new therapeutic armamentarium for the rheumatologist.

## Figures and Tables

**Table 1 jpm-13-01303-t001:** Patient characteristics at baseline.

Patients	120
Females	85.50%
Males	14.50%
Age	62.58 (53–74)
Age ≥ 65	54.6%
Smoke	18.5%
Hormone replacement therapy	1.03%
Previous HZ infection	4.12%
Previous VTE	0%
Previous MACE	7.21%
Previous cancer	0.83%
Diabetes	15.46%
Hypertension	47.42%
Disease duration	7 (4–12)
No biological failure	23.7%
1 biological failure	20.6%
2 biological failures	27.8%
3 biological failures	16.5%
4 biological failures	10.3%
5 biological failures	1.1%
ACPA	80.4%
RF	91.75%
Double positive	79.38%
Methotrexate	35%
Leflunomide	5.15%
Sulfasalazine	4.12%
Statin	12.37%
Average steroid dosage	4 (0–7.7)
DAS28	4.76 (4.34–5.55)
CDAI	21 (18–22.75)
ESR mm/h	35 (18–48)
CRP mg/dl	1.18 (0.5–1.89)
Tender Joints	6 (5–11)
Swollen Joints	4 (4–8)
VAS	7 (5–8)
HAQ	1 (1–1.5)
PGA patient	70 (40–80)
PGA physician	65 (30–70)
Body weight	68 (58–75)
BMI	23.15 (20.79–26.30)
Total cholesterol mg/dL	197 (174–210)
LDL mg/dL	111.40 (101–134)
HDL mg/dl	53 (45–58)
Triglycerides mg/dL	110.5 (90–123.75)
Creatinine mg/dL	0.78 (0.65–0.9)
Aspartate aminotrasferase (AST) UI/L	20 (15–25.25)
Alanine aminotrasferasi (ALT) UI/L	17.5 (14–23)
Hemoglobin g/L	12.55 (11.83–13.22)

**Table 2 jpm-13-01303-t002:** Clinical and laboratory changes during treatment.

	Basal	3 Months	6 Months			
Methotrexate	35%	28.86%	21.6%			
Leflunomide	5.15%	2%	2%			
Sulfasalazine	4.12%	3.10%	3.1%	*p*		
Statin	12.37%	11.34%	13.4%	T0 vs. 3 months	T0 vs. 6 months	3 months vs. 6 months
Average steroid dosage	4 (0–7.7)	0 (0–4)	0 (0–4)	0.0001	0.0001	0.048
DAS28	4.76 (4.34–5.55)	3.03 (2.67–3.63)	2.7 (2.29–3.1)	0.0001	0.0001	0.0001
CDAI	21 (18–22.75)	13 (12–15)	9 (6–10.25)	0.0001	0.0001	0.0001
ESR mm/h	35 (18–48)	15 (10–29.75)	13 (10–28)	0.0001	0.0001	0.026
CRP mg/dL	1.18 (0.5–1.89)	0.47 (02–0.85)	0.28 (0.12–0.49)	0.0001	0.0001	0.0001
Tender Joints	6 (5–11)	2 (1–3)	1 (0–2)	0.0001	0.0001	0.0001
Swollen Joints	4 (4–8)	1 (1–2)	0 (0–1)	0.0001	0.0001	0.0001
VAS	7 (5–8)	2.5 (1–5)	2 (1–3)	0.0001	0.0001	0.0002
HAQ	1 (1–1.5)	0.25 (0–0.5)	0.5 (0–0.5)	0.0001	0.0001	NS
PGA patient	70 (40–80)	27.5 (15–41.25)	20 (10–30)	0.0001	0.0001	0.0001
PGA physician	65 (30–70)	20 (10–45)	20 (10–30)	0.0001	0.0001	NS
Weight	68 (58–75)	65.5 (57.7–73.2)	67 (58–74)	NS	NS	NS
BMI	23.15 (20.79–26.30)	21.97 (20.96–23.12)	22 (20.78–23.81)	NS	NS	NS
Total cholesterol mg/dL	197 (174–210)	189.5 (157–209)	203.5 (174.25–211)	NS	NS	NS
LDL mg/dL	111.40 (101–134)	121 (103.2–134)	121 (103.2–134)	NS	NS	NS
HDL mg/dL	53 (45–58)	58.5 (45–60)	57 (45–61.75)	NS	NS	NS
Triglycerides mg/dL	110.5 (90–123.75)	113.5 (95–121)	113.5 (96.25–130.25)	NS	NS	NS
Creatinine mg/dL	0.78 (0.65–0.9)	0.81 (0.7–0.87)	0.8 (0.62–0.87)	NS	NS	NS
Aspartate aminotrasferase (AST) UI/L	20 (15–25.25)	19 (25–25)	19 (14–23)	NS	0.0033	NS
Alanine aminotrasferasi (ALT) UI/L	17.5 (14–23)	17.5 (12–26)	15 (11.75–22.25)	NS	0.015	NS
Hemoglobin g/L	12.55 (11.83–13.22)	12.6 (12–13.37)	12.64 ± 1.05	NS	NS	NS
MACE	0	0	0			
VTE	0	0	0			
HZ	0	2	0			

## Data Availability

The raw data supporting the conclusions of this article will be made available by the authors, without undue reservation.

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
