# Peer review of "ReLiFiRa (Real Life Filgotinib in Rheumatoid Arthritis): Retrospective Study of Efficacy and Safety in Common Clinical Practice"

_jpm, 2023, doi:10.3390/jpm13091303_

Round 1

Reviewer 1 Report

Descriptive study, showing that Filgotinib in 120 refractory RA patients is effective and safe. Interesting data with some serious limitations. very heterogenous group with patients with 1-5 "biological failures". No control groups and limited information on the previous treatments.

Titel: I would prefer a more message-centered title without the acronym. Almost all disease-dependent values improved after 3 and 6 months, so you can state that in the title.

Please indicate abbreviations in one style. You switch between brackets (e.g. line 1 of the abstract: “(FIL)”) and dashes (e.g. line 8 of the abstract: “-VTE major adverse cardiovascular events-MACE-“).

The aim which is stated at the end of the introduction is not very precise and it is not related to Filgotinib or any other JAKi.

Material and Methods:

It is not described in this section, how the data was gathered. Which tests were used? How were they performed? Inclusion- and Exclusion criteria.

There is more information in the abstract about the nature of the study, the participating centers and the evaluated characteristics than in the M&M section. I would mention the ethical approval in the methods section.

Patient characteristics in my opinion do not belong into the M&M section. I would put the whole “baseline clinicical and demographic characteristic” part into the results section, because you describe unique features of your patient population and nothing you have done or used. Or put the table into an supplemental file, because most of the data is also included in table 2.

Results: The results are clearly presented.

Discussion: Interesting data, but it is difficult to evaluate the actual improvement of this treatment because of the missing control groups. It is great, that Filgotinib can improve the outcome after 1-5 “biological failures”, but to me it is not clear, what that means. Which biologics were previously used? TNF-Inhibitors? Other JAKis like Tofacitinib? It would be good to include data on how many patients used which biologics before the treatment.

Also, can you discuss your data step by step? It is sometimes difficult to follow the discussion and to understand, which observation is in line with the literature and what is new.

Also, please discuss the limitations of the study. Limitations are mentioned in the conclusion, but not discussed before.

Can you state, why you only observed Filgotinib-treated patients? A comparison with other JAKi- and TNF-treated patients would be great.

Good overall quality of english language.

Some minor mistakes I found:

M&M line 1: no comma needed

page 6, first sentence needs improvement

please improve the last sentence in the conclusion

Reviewer 2 Report

The authors collected retrospective data on 120 patients with RA managed across a number of healthcare providers in Northern Italy.

The 120 patients were fairly typical of an RA population. 85% were female, average age 62.58 years, average disease duration7 years; 76% had failed at least one previous biologic, 80% were ACCPA positive and 35% were on methotrexate.

In this ‘real world’ cohort the authors found significant improvement in disease activity at the 3 month timepoint with further improvement at 6 months. There were 2 herpes Zoster episodes reported at the 3 month timepoint but no other significant adverse events.

Minor:

Introduction, line 4 ..event beter.. should be …even better..

Materials and Methods; lines 12 and 15 and 17; replace ‘about’ with ‘regarding’

Round 2

Reviewer 1 Report

Dear authors,

thank you for improving the manuscript! I like it now very much. Really nice data. I have 3 minor issues/suggestions:

Introduction last sentence: I would exchange "data" with "data on safety and efficiency" or modify the sentence accordingly. You can state your aim a bit more confident. You did not just collect "the data". You collected very specifically patient data on safety and efficiency and you can show that Filgotinib is safe and efficient in patients with a refractory disease outcome.

methods line 2: I think you can leave out "in an Excell Database"

discussion: please do not state that it is safe in terms of the occurence of cancer. We cannot say that within an observation time of 6 months.
